# Robust Multi-fidelity Bayesian Optimization with Deep Kernel and Partition

**Fengxue Zhang**
University of Chicago
zhangfx@uchicago.edu

**Thomas A. Desautels**
Lawrence Livermore
National Laboratory
desautels2@llnl.gov

**Yuxin Chen**
University of Chicago
chenyuxin@uchicago.edu

## Abstract

Multi-fidelity Bayesian optimization (MFBO) is a powerful approach that utilizes low-fidelity, cost-effective sources to expedite the exploration and exploitation of a high-fidelity objective function. Existing MFBO methods with theoretical foundations either lack justification for performance improvements over single-fidelity optimization or rely on strong assumptions about the relationships between fidelity sources to construct surrogate models and direct queries to low-fidelity sources. To mitigate the dependency on cross-fidelity assumptions while maintaining the advantages of low-fidelity queries, we introduce a random sampling and partition-based MFBO framework with deep kernel learning. This framework is robust to cross-fidelity model misspecification and explicitly illustrates the benefits of low-fidelity queries. Our results demonstrate that the proposed algorithm effectively manages complex cross-fidelity relationships and efficiently optimizes the target fidelity function.

## 1 Introduction

Multi-fidelity Bayesian optimization (MFBO) (Dai et al., 2019; Wu et al., 2020; Takeno et al., 2020) is increasingly prevalent in the adaptive design of scientific experiments (Buterez et al., 2023), automated hyperparameter optimization (Eggensperger et al., 2021; Pfisterer et al., 2022), and policy optimization in control problems (Letham and Bakshy, 2019; Wu et al., 2020).

Previous work has often relied on various assumptions about the relationship between different fidelities to analyze efficiency theoretically (Song et al., 2019; Kandasamy et al., 2016, 2017). Similar to transfer learning for Bayesian optimization (BO), a more practical challenge involves handling significant misalignment between fidelities while maintaining cost efficiency. Recent approaches to robust transfer learning for BO (Appice et al., 2015; Probst et al., 2019; Perrone et al., 2019; Reif et al., 2012; Pfisterer et al., 2021; Feurer et al., 2018) and robust single-fidelity BO against model misspecification (Bogunovic and Krause, 2021; Liu et al., 2023) have addressed this issue yet typically do not consider the sample efficiency on the lower fidelities. Some research suggests mitigating the problem by avoiding evaluations or learning from unreliable low-fidelity sources (Mikkola et al., 2023; Foumani et al., 2023), but they do not explicitly deal with errors incurred from the unreliable model learning of the multi-fidelity structure in the model design and acquisition.

Leveraging recent advancements in efficient kernel learning, uncertainty quantification, and error bounds for learning algorithms (Xu and Raginsky, 2017; Robinson et al., 2020; Wang et al., 2021), we propose a general-purpose framework that uses sampling-based cost-aware acquisition. This framework captures complex and potentially misaligned multi-fidelity evaluations while explicitly addressing model misspecification on the fly with robust data acquisition and deep kernel learning.

Workshop on Bayesian Decision-making and Uncertainty, 38th Conference on Neural Information Processing Systems (NeurIPS 2024).

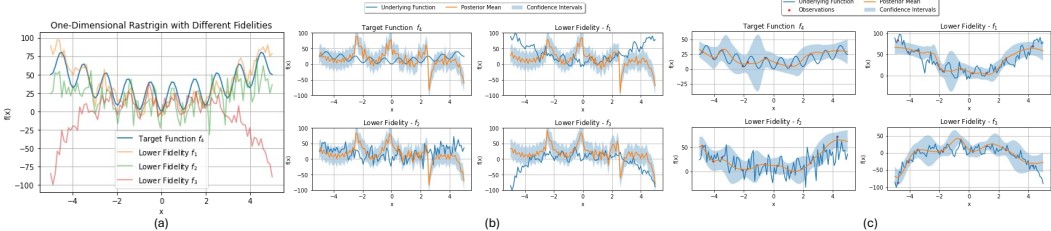

Figure 1: We illustrate the problem of learning and optimization on both target fidelity and misaligned low fidelities. (a) demonstrates the one-dimensional Rastrigin function (Pohlheim, 2007) and the manually constructed low fidelities. (b) demonstrates the posterior of learning the multi-fidelity functions with conventional multi-task GP (Swersky et al., 2013) previously applied in MFBO (Letham and Bakshy, 2019) when feeding 2000 training points densely distributed in the search space. (c) shows the posterior of the proposed model when feeding 10 points from each fidelity.

## 2  Preliminaries

We begin by introducing useful notation, mostly following previous work by Song et al. (2019), and formally stating the problem studied in this paper.

### 2.1  Multi-fidelity Optimization of Unknown Objective

Consider the problem of maximizing an unknown payoff function $f_M : \mathbf{X} \to \mathbb{R}$. We can probe this function by directly querying it at some point $\mathbf{x} \in \mathbf{X}$, consequently obtaining a noise-free observation $y_{\langle \mathbf{x}, M \rangle} = f_M(\mathbf{x})$. In addition to $f_M$, we have access to oracle calls for unknown auxiliary functions $f_1, \ldots, f_{M-1} : \mathbf{X} \to \mathbb{R}$. Similarly, querying any $f_\ell$ at $\mathbf{x}$ yields a noise-free observation $y_t = f_{\ell_t}(\mathbf{x}_t)$. Each auxiliary function $f_\ell$ can be viewed as a lower-fidelity version of $f_M$ when $\ell < M$. Specifically, we model the unknown target fidelity functions with corresponding Gaussian process (GP): $f_M \sim \mathrm{GP}\left(\mu_M(\mathbf{x}), k_M(\mathbf{x}, \mathbf{x}')\right)$, where $\mu_M$ and $k_M$ denote the prior mean and covariance. Let $\langle \mathbf{x}, \ell \rangle$ denote the action of querying $f_\ell$ at $\mathbf{x}$. Each action $\langle \mathbf{x}, \ell \rangle$ incurs a cost of $\lambda_\ell$ and yields a reward:

$$r(\langle \mathbf{x}, \ell \rangle) = \begin{cases} f_M(\mathbf{x}) & \text{if } \ell = M \\ r_{\min} & \text{otherwise} \end{cases}$$

That is, performing $\langle \mathbf{x}, M \rangle$ at the target fidelity level achieves a reward $f_M(\mathbf{x})$. The collective historical observations after $T$ iteration is denoted by $\mathcal{D}_T \triangleq \{\langle \mathbf{x}_t, \ell_t \rangle, y_t\}_{t=1\ldots T}$. We also define the collective historical observations up to certain fidelity $\ell$ as $\mathcal{D}_{\ell,T} \triangleq \{\langle \mathbf{x}_t, \ell'_t \rangle, y_t\}_{1 \le t \le T, \ell'_t \le \ell}$. When given a fixed budget, we need to guarantee the cumulative cost does not exceed the budget, i.e., $\sum_{t=1}^{T} \lambda_{\ell_t} \le \Lambda$. Lower fidelity actions $\langle \mathbf{x}, \ell \rangle$ for $\ell < M$ yield the minimal immediate reward $r_{\min}$ but can provide valuable information about $f_M$, potentially leading to better decisions later. Without loss of generality, we assume $\max_{\mathbf{x}} f_M(\mathbf{x}) \ge 0$ and $r_{\min} \equiv 0$. Note that we define the reward only as incurred based on the target fidelity, and the query on low fidelity does not incur a reward but only helps with the learning. Hence, in the context of multi-fidelity Bayesian optimization, the **simple regret (SR)** is defined as: $\mathbf{R}(\hat{\mathbf{x}}) = f_M(\mathbf{x}^*) - f_M(\hat{\mathbf{x}})$, where $\hat{\mathbf{x}} := \arg\max_{\mathbf{x}:(\langle \mathbf{x}, M \rangle, y) \in \mathcal{D}_T} f_M(\mathbf{x})$ is a point selected to be evaluated at the target fidelity, and $\mathbf{x}^*$ is the global maximizer of the function $f_M$. Our objective is to find the candidate that minimizes the simple regret after **exhausting a given budget** $\Lambda$.

### 2.2  Expected Excess Risk

In addition to the conventional analysis that assumes the prior is properly specified, we explicitly deal with the model misspecification regarding the difference between the posterior mean and true underlying function. In the context of statistical learning theory, the **convergence rate** of the expected excessive risk with respect to the training dataset $\mathcal{D}_{\ell,T}$ at fidelity $\ell \in [M]^+ = [1 \ldots M]$ after $T$ iterations is defined as $\mathrm{Rate}_{MF}(\ell, T) \triangleq \mathbb{E}\left[\mathcal{L}(f_\ell, \tilde{f}_\ell) | \mathcal{D}_{\ell,T}\right] - \mathcal{L}(f_\ell, \tilde{f}_\ell^*) = O(T^\alpha)$, Here, $-1 < \alpha < 0$ is a constant that characterizes the rate of convergence. $\tilde{f}_\ell$ is the hypothesis of fidelity $\ell$ produced by the learning algorithm when trained on $\mathcal{D}_{\ell,T}$. $\mathcal{L}(f, \tilde{f})$ is the loss function evaluating the hypothesis $\tilde{f}$ with respect to the true objective $f$, and $\tilde{f}_\ell^*$ is the hypothesis on fidelity $\ell$ that minimizes the expected loss.

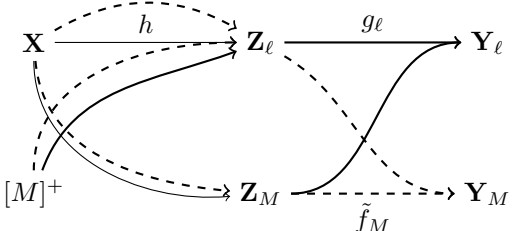

Figure 2: Schema for multi-fidelity learning implemented with deep kernel. The dotted lines denote the flow of target fidelity (strong data), and the solid lines the flow of low fidelities(weak data). Here, we denote the output space for certain fidelity $\ell \in [M]^+$ as $\mathbf{Y}_\ell$. Specifically, the target fidelity output space is denoted as $\mathbf{Y}_M$.

# 3 Method

In this section, we discuss the model design and the analysis-inspired data acquisition procedure of the proposed **R**obust **M**ulti-**F**idelity **B**ayesian **O**ptimization with **D**eep Kernel Learning and **P**artition (RMFBO-DP). The pseudo code and detailed design choices are deferred to Appendix B.

## 3.1 Model

We employ hierarchical deep kernel learning (Wilson et al., 2016) regularized with the spectral norm (Van Amersfoort et al., 2021) with mean absolute error (MAE) as the loss function to deal with overfitting caused by shortage of training data and the specific choice of training loss. We assume that the multiple fidelities $\{f_\ell\}_{\ell \in [M]}$ are mutually dependent through some fixed, possibly unknown joint probability distribution. Therefore, we seek to approximate the joint underlying function $f : \mathbf{X} \times [M]^+ \to \mathbb{R}$, where both the position $\mathbf{x} \in \mathbf{X}$ and fidelity $\ell \in [M]^+$ are inputs, and learn the approximation $\tilde{f}$ through joint learning of $h$ and $g_{\ell \in [M]^+}$. Here, a *single* latent space mapping $h : \mathbf{X} \times [M]^+ \to \mathbf{Z}$ convert the input space $\mathbf{X}$ to the latent space $\mathbf{Z}$ which consists of the fidelity independent part $\mathbf{Z}_M$ and fidelity dependent part $\mathbf{Z}_\ell$. On top of the latent space, we construct a set of objective mappings $g_1 \ldots g_M$ for each fidelity. Namely $\forall \ell \in [M]^+$, $f_\ell \triangleq f(\cdot, \ell)$ and $f(\cdot, \ell)$ is approximated by $\tilde{f}_\ell \triangleq g_\ell(h(\cdot, \ell))$. We illustrate the model structure in figure 2. The model is trained by first traversing all low fidelities for weak learning, then jointly optimizing the deep kernel.

## 3.2 Data Acquisition

We extend random exploration to the multi-fidelity regime. We use both the expected generalization error and SR bounds to guide the cost-efficient acquisition. When the error bound contributes more to global regret, we randomly explore until it is more cost-efficient to conduct target fidelity acquisition. When the SR contributes more to the general regret, we conduct target fidelity acquisition on certain partitions of the search space. We leverage the GP posterior calibrated with the excess risk on observed points to exclude from acquisition the partitions of the search space that, with high probability, do not contain the global optimum. To do so, we rely on both the upper confidence bound $\text{UCB}_{f_M,t}(\mathbf{x}) \triangleq \mu_{f_M,t-1}(\mathbf{x}) + \beta_{f_M,t}^{1/2}\sigma_{f_M,t-1}(\mathbf{x})$ and lower confidence bound $\text{LCB}_{f_M,t}(\mathbf{x}) \triangleq \mu_{f_M,t-1}(\mathbf{x}) - \beta_t^{1/2}\sigma_{f_M,t-1}(\mathbf{x})$, where $\sigma_{t-1}(\mathbf{x}) = k_{t-1}(\mathbf{x},\mathbf{x})^{1/2}$ and $\beta_t$ acts as a scaling factor corresponding to certain confidence. Formally, the acting search space at iteration $t$ is $\hat{\mathbf{X}}_t = \{x \in \mathbf{X} \mid \text{UCB}_{f_M,t}(\mathbf{x}) > \max_{\mathbf{x}' \in \mathbf{X}} \text{LCB}_{f_M,t}(\mathbf{x}') - \text{Rate}_{MF}(M,t)\}$.

Regarding the theoretical justification, previous works offer an upper bound for cumulative regret when applying random exploration on the partitions of interest (Salgia et al., 2024) defined as $\{x \in \mathbf{X} \mid \text{UCB}_{f_M,t}(\mathbf{x}) > \max_{\mathbf{x}' \in \mathbf{X}} \text{LCB}_{f_M,t}(\mathbf{x}')\}$, as the target fidelity acquisition function when we ignore the contribution of low-fidelity evaluation to the target fidelity learning. We extend the cumulative regret bound into the following form of SR bound.

**Informal Theorem 1** *Under proper assumptions and choices of parameters, when ignoring the contribution of performance by learning on the low fidelities, we have with probability at least*

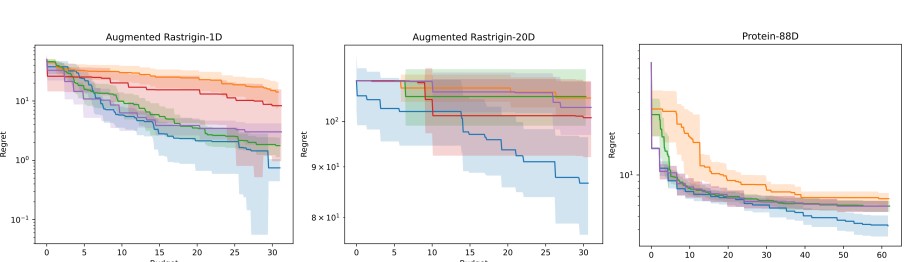

Figure 3: We illustrate the performance of RMFBO-DP compared against MF-MES, MF-KG, rMF-MES, and rMF-KG on both synthetic and real-world datasets. The results are collected from 10 independent trials. The y-axis denotes the simple regret, and the x-axis denotes the consumed budget. The shades area shows the $95\%$ confidence interval. We trim the shared initial single data point. Detailed discussions are deferred to Appendix C.

$1 - \delta$, $SR(t) = \tilde{\mathcal{O}}\left(\sqrt{\frac{\gamma_{T_M(t)}}{T_M(t)}} \log \frac{T_M(t)}{\delta}\right)$. Here $\tilde{\mathcal{O}}$ means up to the logarithmic factor, and $T_\ell(t) \triangleq |\{\langle \mathbf{x}_{t'}, \ell_{t'}\rangle, y_{t'}\}_{1 \leq t' \leq t, \ell_{t'}=\ell}|$ denotes evaluations at fidelity $\ell$ among $t$ evaluations.

We exploit recent advancements in expected excess risk in meta-learning (Robinson et al., 2020) to extend the previous SR results to multiple weak learning sources. We state the informal version of the theoretical results here while deferring the discussion of assumptions, proof, and other details to Appendix A. First, we decompose the ultimate SR into separate components for the conventional regret and the generalization error.

**Informal Theorem 2** *The misspecification-aware simple regret ($SR_{MA}$) of the proposed algorithm can be decomposed into the standard simple regret (SR) and the rate term as $SR_{MA}(t) \leq SR(t) + Rate_{MF}(M, t)$.*

In the following, we generalize the meta-learning expected excess risk (Robinson et al., 2020; Xu and Raginsky, 2017) to multi-fidelity learning.

**Informal Theorem 3** *When the lowest single fidelity bears the convergence rate $Rate_{MF}(1, T) = O(\sqrt{\frac{1}{T_1}})$, the excess risk bound $Rate_{MF}(\ell, t) \leq$*

$$O\left(Rate_{MF}(\ell-1, t) + \frac{\sqrt{\left(\log_{T_\ell(t)} Rate_{MF}(\ell-1, t) + 1\right)\log T_\ell(t)}}{T_\ell(t)}\right)$$

This allows us to differentiate the multiple fidelities' contribution to the target fidelity learning and regret minimization. Specifically, a cost-aware multi-fidelity acquisition could be made by minimizing the $SR_{MA}(T)$ such that the total cost incurred by querying different fidelities does not exceed $\Lambda$.

### 3.3 Evaluation

We evaluated the proposed algorithm RMFBO-DP against four baselines on both synthetic datasets corresponding to figure 1 and real-world protein design dataset. The results shown in figure 3 demonstrate the robustness of RMFBO-DP in various tasks and efficiency in tasks of different dimensionalities. We defer detailed description to Appendix C.

## 4 Conclusion

In this paper, we introduced a novel multi-fidelity Bayesian optimization approach focusing on robustness and efficiency. Our method explicitly addresses the misspecification issues in multi-fidelity deep model learning by incorporating budget-sensitive low-fidelity sampling and constraining acquisitions to a subset of the global search space for target fidelity optimization. By tackling the challenges of low-fidelity misalignment and efficient target fidelity optimization in a principled, cost-effective manner, we demonstrated that our approach significantly improves robustness and performance over existing methods, as confirmed by our theoretical and empirical results.

## Acknowledgements

This work was performed under the auspices of the U.S. Department of Energy by Lawrence Livermore National Laboratory under Contract DE-AC52-07NA27344. LLNL-CONF-867786. The GUIDE program is executed by the Joint Program Executive Office for Chemical, Biological, Radiological and Nuclear Defense (JPEO-CBRND) Joint Project Lead for CBRND Enabling Biotechnologies (JPL CBRND EB) on behalf of the Department of Defense's Chemical and Biological Defense Program. This effort was in collaboration with the Defense Health Agency (DHA) COVID funding initiative. The views expressed in this paper reflect the views of the authors and do not necessarily reflect the position of the Department of the Army, Department of Defense, nor the United States Government. References to non-federal entities do not constitute nor imply Department of Defense or Army endorsement of any company or organization. This work was completed in part with resources provided by the University of Chicago's Research Computing Center.

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

## A   Theorems and Proofs

We begin by introducing the necessary notion.

**Definition 1 (RKHS Salgia et al. (2024))** *Consider a positive definite kernel $k : \mathcal{X} \times \mathcal{X} \to \mathbb{R}$. A Hilbert space $\mathcal{H}_k$ of functions on $\mathcal{X}$ equipped with an inner product $\langle \cdot, \cdot \rangle_{\mathcal{H}_k}$ is called a Reproducing Kernel Hilbert Space (RKHS) with reproducing kernel $k$ if the following conditions are satisfied: (i) $\forall x \in \mathcal{X}, k(\cdot, x) \in \mathcal{H}_k$; (ii) $\forall x \in \mathcal{X}, \forall f \in \mathcal{H}_k, f(x) = \langle f, k(\cdot, x) \rangle_{\mathcal{H}_k}$. For simplicity, we use $\psi_x$ to denote $k(\cdot, x)$. The inner product induces the RKHS norm, $\|f\|_{\mathcal{H}_k}^2 = \langle f, f \rangle_{\mathcal{H}_k}$.*

The following discusses the necessary assumptions, formal theorems, and corresponding proofs. We state the typical assumption for BO performance analysis first.

**Assumption 1** *Throughout the optimization, $\tilde{f}_M$ bears an upper bound $B$ on the RKHS norm (Salgia et al., 2024) corresponding to the learned deep kernels $k_t$. Namely $\left\| \tilde{f}_M \right\|_{\mathcal{H}_{k_t}} \leq B$.*

**Assumption 2** *(Generalized Assumption 4.1 of Salgia et al. (2024)) For all $n \in \mathbb{N}$, there exists a discretization $\mathcal{D}_n$ of $\mathcal{X}$ such that for all $f \in \mathcal{H}_k$,*

$$|f(\mathbf{x}) - f([\mathbf{x}]_{\mathcal{D}_n})| \leq \|f\|_{\mathcal{H}_k}/n \quad and \quad |\mathcal{D}_n| = poly(n)^3,$$

*where $[x]_{\mathcal{D}_n} = \arg\min_{y \in \mathcal{D}_n} \|x - y\|_2$, is the point in $\mathcal{D}_n$ that is closest to $x$.*

**Assumption 3** *(Assumption 4.2 of Salgia et al. (2024)) Let $\mathcal{L}_\eta = \{x \in \mathcal{X} \mid f(x) \geq \eta\}$ denote the level set of $f$ for $\eta \in [-B, B]$. We assume that for all $\eta \in [-B, B]$, $\mathcal{L}_\eta$ is a disjoint union of at most $M_f < \infty$ components, each of which is closed and connected. Moreover, for each such component, there exists a bi-Lipschitzian map[1] between each such component and $\mathcal{X}$ with normalized Lipschitz constant pair $L_f, L'_f < \infty$.*

We then state the assumption for analysis of the statistical learning.

**Assumption 4** *The hypothesis space for each fidelity $\mathcal{H}_\ell$ contains the underlying functions $f_\ell$ for $\forall \ell \in [M]^+$.*

**Assumption 5** *We assume the $\forall \ell \in \{2, \ldots, M\}$, $\tilde{f}_\ell$ is L-Lipschitz relative to the function space $\mathcal{H}_h$, meaning $\forall \mathbf{x} \in \mathbf{X}$, $\forall y \in \mathbf{Y}_\ell$, $\forall h, h' \in \mathcal{H}_h$, we have $|\mathcal{L}(y, g_\ell(h(\mathbf{x}))) - \mathcal{L}(y, g_\ell(h'(\mathbf{x})))| \leq L\mathcal{L}(g_{\ell-1}(h'(\mathbf{x})), g_{\ell-1}(h'(\mathbf{x})))$.*

This generalizes the assumption of Theorem 10 from Robinson et al. (2020).

In the following section, we restate the key insights for this work.

---

[1]A map $f : X \to Y$ is bi-Lipschitzian if there exist constants $c_1, c_2 > 0$ such that $c_1 \|x - y\| \leq \|f(x) - f(y)\| \leq c_2 \|x - y\|$ for all $x, y \in X$.

**Theorem 1** *When Assumptions 1, 2, and 3 hold, constraining the random exploration on the target fidelity on $\{x \in \mathbf{X} \mid \mathrm{UCB}_{f_M,t}(\mathbf{x}) > \max_{\mathbf{x}' \in \mathbf{X}} LCB_{f_M,t}(\mathbf{x}')\}$, and choosing $\beta = B^2$, we have the following bound with probability at least $1 - \delta$,*

$$SR(t) = \tilde{\mathcal{O}}(\sqrt{\frac{\gamma_{T_M(t)}}{T_M(t)}} log(T_M(t)/\delta)) \tag{1}$$

*Proof:* A direct extension of Theorem 4.3 and Theorem 4.5 of Salgia et al. (2024) by expanding the noise-free bound in its Theorem 4.3 to the noisy scenario bound in Theorem 4.5 is that when only considering single fidelity optimization applying random exploration in the region of interest defined above, with probability at least $1 - \delta$, the cumulative regret bears the upper bound $\tilde{\mathcal{O}}(\sqrt{\gamma_{T_M(t)} T_M(t)} log(T_M(t)/\delta))$. Adding $T_M(t)$ to the denominator converts it to a high probability upper bound of SR. $\qquad\square$

**Theorem 2** *Under the assumptions of Theorem 1 except for constraining random exploration in $\hat{\mathbf{X}}_t$ as defined in equation 3. The misspecification-aware Bayesian simple regret ($SR_{MA}$) of the proposed algorithm can be decomposed into the standard Bayesian simple regret (SR) and the rate term as follows:*

$$SR_{MA}(t) \leq SR(t) + Rate_{MF}(M, t).$$

*Proof:* The result is a simple extension as we extend the region of interest to $\hat{\mathbf{X}}_t$ as defined in equation 3. The additional term $Rate_{MF}(M, t)$ count for the introduced excess risk. $\qquad\square$

**Theorem 3** *(Generalized Theorem 10 of Robinson et al. (2020)) With the aforementioned assumptions 1-5 hold, and the lowest single fidelity bears the convergence rate $Rate_{MF}(1, T) = O(\sqrt{\frac{1}{T_1}})$. The excessive risk bears the bound*

$$Rate_{MF}(\ell, t) \leq O\left(Rate_{MF}(\ell - 1, t) + \frac{\sqrt{\left(\log_{T_\ell(t)} Rate_{MF}(\ell - 1, t) + 1\right)\log T_\ell(t)}}{T_\ell(t)}\right) \tag{2}$$

*Proof:* With aforementioned assumptions hold, for $\forall 1 < \ell \leq M$, the learning on fidelity $\ell - 1$ and $\ell$ meets the assumption of Theorem 10 of Robinson et al. (2020), then the bound above could be direct results of recursively applying the theorem 10 for $\ell = 2 \ldots M$. $\qquad\square$

# B  Additional Algorithm Details

In the following, we offer the additional details of implementing the proposed algorithm RMFBO-DP.

**Reliable search space exclusion**  We leverage the GP posterior calibrated with the excess risk on observed points to exclude from acquisition the regions that, with high probability, do not contain the global optimum. To do so, we rely on both the upper confidence bound $\mathrm{UCB}_{f_M,t}(\mathbf{x}) \triangleq \mu_{f_M,t-1}(\mathbf{x}) + \beta_{f_M,t}^{1/2}\sigma_{f_M,t-1}(\mathbf{x})$ and lower confidence bound $\mathrm{LCB}_{f_M,t}(\mathbf{x}) \triangleq \mu_{f_M,t-1}(\mathbf{x}) - \beta_t^{1/2}\sigma_{f_M,t-1}(\mathbf{x})$, where $\beta_t$ is the scaling factor corresponding to certain confidence. Formally, the acting search space at iteration $t$ is $\hat{\mathbf{X}}_t$ defined as

$$\left\{x \in \mathbf{X} \mid \mathrm{UCB}_{f_M,t}(\mathbf{x}) > \max_{\mathbf{x}' \in \mathbf{X}} \mathrm{LCB}'_{f_M,t}(\mathbf{x}')\right\}. \tag{3}$$

Here $\mathrm{LCB}'_{f_M,t}(\mathbf{x}') \triangleq \mathrm{LCB}_{f_M,t}(\mathbf{x}') - Rate_{MF}(M, t)$ generalize the lower confidence bound to the incorporate the expected generalization error.

**Algorithm 1** Robust Multi-Fidelity Bayesian Optimization with Deep Kernel Learning and Partition (RMFBO-DP)

---

1: **Input**:Search space $\mathbf{X}$, $t_1$ initial observation $\mathcal{D}_{t_1}$, total budget $\Lambda$;
2: Initialize timestamp as $t \leftarrow t_1$.
3: Initialize remaining budget $\Lambda \leftarrow \Lambda - \sum_{t'=1}^{t} \lambda_{\ell_{t'}}$
4: **while** $\Lambda > 0$ **do**
5:     Update the model $\tilde{f}$.
6:     Identify ROIs $\hat{\mathbf{X}}_t$ according to equation 3.
7:     Maximize cost-sensitive $\mathrm{SR}_{\mathrm{MA}}(t+1) - \mathrm{SR}_{\mathrm{MA}}(t)$ reduction.
        $\Delta_{\mathrm{SR},t} \leftarrow \frac{\mathrm{SR}(t+1)-\mathrm{SR}(t)}{\lambda_M}$
        $\Delta_{\mathrm{Rate}_{MF},t} \leftarrow \max_{\ell \in [M]^+} \frac{\mathrm{Rate}_{MF}(\ell,t+1)-\mathrm{Rate}_{MF}(\ell,t)}{\lambda_\ell}$
8:     **if** $\Delta_{\mathrm{SR},t} \geq \Delta_{\mathrm{Rate}_{MF},t}$ **then**
9:         Draw random sample $\mathbf{x}_t$ from $\hat{\mathbf{X}}_t$ on fidelity $\ell_t$
        $\ell_t \leftarrow M$
        $\mathbf{x}_t \leftarrow \max_{\mathbf{x} \in \hat{\mathbf{X}}} \alpha_{f_M}(\mathbf{x})$
10:     **else**
11:         Sample candidate on on fidelity $\ell_t$
        $\ell_t \leftarrow \arg\max_{\ell \in [M]^+} \frac{\mathrm{Rate}_{MF}(\ell,t+1)-\mathrm{Rate}_{MF}(\ell,t)}{\lambda_\ell}$
        $\mathbf{x}_t \leftarrow$ sample from uniform distribution on $\mathbf{X}$.
12:     **end if**
13:     Update the observation $\mathcal{D}_{t+1} \leftarrow \mathcal{D}_t \cup \{\langle \mathbf{x}_t, \ell'_t \rangle, y_t\}$
14:     Update the remaining budget $\Lambda \leftarrow \Lambda - \lambda_\ell$
15:     Update the timestamp $t \leftarrow t + 1$
16: **end while**

---

**Estimation of SR rates**   Due to the difficulty of analyzing exact SR, we rely on the following simple approximation. For two consecutive evaluations of the target fidelity, if we observe improvement in the best reward, we leverage the improvement to regress the SR. Namely, for $\forall 1 \leq t_1 < t_2 \leq T$, if $\Delta_{f_M} \triangleq y_{\langle \mathbf{x}_{t_2}, \ell_t = M \rangle} - y_{\langle \mathbf{x}_{t_1}, \ell_t = M \rangle} > 0$, we update the approximation for $\mathrm{SR}(t_2)$ by solving $\Delta_{f_M} = \mathrm{SR}(t_2) - \mathrm{SR}(t_1)$.

**Estimation of expected excess risk**   Similar to the above approximation of SR, we approximate the excess risk reduction by regressing to the observed fitting error improvement. $\mathrm{Rate}_{MF}(M, t)$. for $\forall 1 \leq t_1 < t_2 \leq T, \ell \in [M]^+$, we resort to 5-fold cross-validation on $\mathcal{D}_{\ell, t_1}$ and $\mathcal{D}_{\ell, t_2}$ to estimate the model fitting improvement $\Delta_{\mathcal{L}(f_\ell, \tilde{f}_\ell), t} = \mathrm{Rate}_{MF}(\ell, t_2) - \mathrm{Rate}_{MF}(\ell, t_1)$. Solving the equation allows us to approximate $\mathrm{Rate}_{MF}(\ell, t_2)$.

**Constraining acquisition**   We rely on random discretization to constrain the acquisition within $\hat{\mathbf{X}}_t$, which rejects the candidates outside $\hat{\mathbf{X}}_t$. Note popular BO frameworks typically allow optimizing the acquisition function subject to constraints, e.g., Botorch (Balandat et al., 2020).

# C   Experiments

We compare the proposed RMFBO-DP against four baselines. The first one is the entropy-based method denoted as MF-MES proposed by Takeno et al. (2020); the second one is denoted as MF-KG, which is the cost-efficient knowledge gradient method proposed by Wu et al. (2020). The third and fourth algorithms are corresponding variants when applying the robust MFBO framework proposed by Mikkola et al. (2023), denoted as rMF-MES and rMF-KG correspondingly. We rely on BoTorch (Balandat et al., 2020) and gpytorch (Gardner et al., 2018) to implement RMFBO-DP and the baselines.

## C.1  Dataset

We evaluate algorithm performance on synthetic datasets and a real-world multi-fidelity protein design task.

**Rastrigin dataset**  As illustrated in figure 1, we construct the four-fidelity version of Rastrigin function (Pohlheim, 2007) on 1D search space. We further extend the construction to 20D search space. Here, the first two lower fidelities generally share the same trend as the target-fidelity underlying function, while the lowest fidelity that incurs the cheapest evaluation cost disagrees with the target fidelity function except for limited central area and largely diverges in the border areas.

**Multi-fidelity protein design**  We use a protein engineering dataset describing a set of antigen/antibody binding calculations. These calculations, executed using supercomputing resources, estimate the change in binding free energy at the interface between each of the 71769 modified antibodies and the SARS-CoV-2 spike protein, as compared to the single reference antibody from which they are derived. Estimations of binding free energy ($\Delta\Delta G$) are calculated using protein-structure-based Rosetta Flex simulation software (Das and Baker, 2008; Barlow et al., 2018) and FoldX (Schymkowitz et al., 2005; Sapozhnikov et al., 2023; Buß et al., 2018). We treat Rosetta's outcomes as the objective of the target fidelity. These calculations took several CPU hours each and were produced during an antibody design process (Desautels et al., 2020, 2022).

## C.2  Evaluation

We evaluated the proposed algorithm Robust Multi-fidelity Bayesian Optimization (RMFBO-DP) against four baselines on both synthetic datasets corresponding to figure 1 and four real-world tasks. We've shown that the proposed algorithm outperforms the baselines in terms of simple regret on Rastrigin-1D, Rastrgin 20d, and Protein-88D.

## C.3  Ablation Study

We conduct an ablation study to investigate the impact of the proposed algorithm components. We compare the performance of the additional variants of the baselines when applying the same deep kernel learning yet without random sampling on low-fidelities. As is shown in table 1, the performance of the variants is not consistently improved upon the corresponding baselines and lags behind RMFBO-DP. This observation suggests that data acquisition is crucial to the performance improvement of the proposed RMFBO-DP.

| Method | Rastrigin-1D | Rastrigin-20D |
|---|---|---|
| **RMFBO-DP** | **0.75 ± 0.30** | **86.64 ± 9.80** |
| MF-MES | 15.00 ± 6.83 | 105.66 ± 4.25 |
| MF-KG | 1.84 ± 0.61 | 106.03 ± 6.96 |
| MF-MES-DK | 2.83 ± 2.57 | 106.62 ± 5.72 |
| MF-KG-DK | 7.04 ± 4.60 | 110.00 ± 0.00 |
| rMF-MES | 8.33 ± 7.36 | 100.93 ± 8.87 |
| rMF-KG | 3.02 ± 1.18 | 103.37 ± 6.41 |
| rMF-MES-DK | 14.88 ± 7.65 | 103.20 ± 6.24 |
| rMF-KG-DK | 3.37 ± 1.23 | 103.52 ± 5.32 |
| Random | 5.05 ± 2.43 | 104.06 ± 6.15 |

Table 1: Ultimate simple regrets for Rastrigin-1D and Rastrigin-20D tasks. The best-performing algorithm for each task is highlighted in bold. The results are collected from at least ten independent trials. We mark the variants using a deep kernel with "-DK".

