# OpenReview forum: "Robust Multi-fidelity Bayesian Optimization with Deep Kernel and Partition"
_NeurIPS.cc/2024/Workshop/BDU — NeurIPS BDU Workshop 2024 Poster_

### Official Review · Reviewer_1SMq · 2024-09-26
**An interesting paper but perhaps lacking in justification and detail**

**Rating:** 5
**Confidence:** 3

**Review:**

**Overview**: The paper demonstrates a new method for multi fidelity Bayesian optimisation using a hierarchical deep kernel and random exploration of a search space constrained by the UCB, LCB, and the expected excess risk (defined as the expected difference in loss between the estimated and optimal hypothesis function at each fidelity level). Analysis of the simple regret of this approach is presented but the implications are not discussed in detail. Experimental results seem promising but it is hard to identify whether the selected baselines are comparable (eg how much of the performance improvement is due to is the deep kernel vs the new excess risk term in the search space).

**Quality**: The underlying work seems of a reasonably high standard. The core idea of the method (explicitly accounting for the misspecification at each fidelity) is sound and deserving of study. Empirically, the proposed algorithm performs very well. However, the paper itself is missing detail that would enable the reader to connect a) their method to existing methods and b) the elements of their method to the observed improvements (both in theory and in practice). In particular, the reason for combining their search space that accounts for "excess risk" (to my knowledge a novel component) with a deep kernel is not explained, which makes it hard to assess the quality of their new contribution.

**Clarity**: The text of the paper is mostly well written. The multi-fidelity problem is introduced well. The visualisations of the posteriors of different methods is insightful, but could benefit from larger font sizes and additional discussion. The visualisation of the model architecture and corresponding explanation is in my opinion insufficient to understand the implementation or reason behind it; however I have not had experience with deep/hierarchical kernel learning so it may be due to a lack of expertise on my part. Notation is occasionally introduced without being concretely defined (what loss function is used? Where is the evaluation budget $\Lambda$ used?). The algorithm is introduced in an appendix (probably due to space constraints) but is not discussed or explained in any detail. Many of the proofs are sketches/outlines rather than step-by-step which, while good for readability, makes it hard to assess the validity; however, as noted by the authors and in the next section this may because they are fairly small contributions. The title of the paper is in my opinion slightly incorrect English, and 'Partition' is not explained (presumably refers to the regions-of-interest search scheme). The introduction is good and the literature review sufficient, but the analysis/discussion of the method is insufficient to appreciate the contributions of this work.

**Originality and significance**: The authors' method for explicitly accounting for misspecification in multi fidelity problems is as far as I know an original idea in Bayesian optimisation. The theory seems a simple extension to existing work, which is acknowledged by the authors in the proofs ("direct extension", "simple extension", "direct result of applying"). The results seem promising but could benefit from being better discussed and analysed so that the significance and applicability of the ideas in their algorithm can be appreciated.

**Pros and cons**:

*Pros*
- Generally well written
- Good experimental results
- Good introduction of the problem
- Great underlying idea and approach to tackling the problem
- Idea is backed up with theoretical analysis

*Cons*
- What "Partition" refers to is not explained
- Cost-aware nature of the algorithm not explained
- Impact of hierarchical kernel vs new search scheme not identified or discussed
- (Opinion) Model architecture insufficiently explained
- Analysis and discussion somewhat lacking

---

### Official Review · Reviewer_AQ7t · 2024-10-02
**Robust Multi-fidelity Bayesian Optimization with Deep Kernel and Partition**

**Rating:** 6
**Confidence:** 3

**Review:**

Pros:
- The paper presents a reasonable approach to multi-fidelity optimization that relies on recent theoretical results
- Some experimental evidence suggests the superiority of the proposed approach

Cons:
- The approach is a straightforward application of the results by Salgia et al. (2024), making it an extension of that work
- Given the first cone, deeper experimental evidence would make a paper a solid work, while the main text has a single paragraph without information on used baselines, and all relevant information moved to the appendix.

Misprints and presentation problems:
1. What is the assumption about f_l for l < M? I see only the GP prior for f_M in Section 2.1
2. The convergence rate should be of the order O(T^l), not O(t^l)
3. For Figure 2, please make arrows parallel to each other and not overly, as it is hard to spot now the edges that have both sold and dashed markup.
4. k_{t - 1} is never defined
5. Put a more transparent link to the algorithm RFBO in the main text.

---

### Decision · Program_Chairs · 2024-10-09

**Decision:**

Accept (Poster)

**Comment:**

Reviews are borderline, but the actual text indicates a more positive reception than what is seen in the scores. The main weaknesses listed involve comprehensiveness. This is okay, given this is a workshop submission rather than a conference paper. Overall, I am of the impression the work lands on the "accept" side of the cutoff for borderline papers.